# A Two-Stage DEA Approach to Measure Operational Efficiency in Vietnam's Port Industry

Chia-Nan Wang [1], Phi-Hung Nguyen [2,3,*], Thi-Ly Nguyen [1,*], Thi-Giang Nguyen [3], Duc-Thinh Nguyen [3], Thi-Hoai Tran [3], Hong-Cham Le [3] and Huong-Thuy Phung [3]

[1] Department of Industrial Engineering and Management, National Kaohsiung University of Science and Technology, Kaohsiung 80778, Taiwan; cn.wang@nkust.edu.tw

[2] Department of Business Management, National Taipei University of Technology, Taipei 10608, Taiwan

[3] Faculty of Business, FPT University, Hanoi 100000, Vietnam; giangnths140548@fpt.edu.vn (T.-G.N.); thinhndhs140747@fpt.edu.vn (D.-T.N.); hoaitths140547@fpt.edu.vn (T.-H.T.); chamlthhs140430@fpt.edu.vn (H.-C.L.); thuyphhs140590@fpt.edu.vn (H.-T.P.)

[*] Correspondence: hungnp30@fe.edu.vn (P.-H.N.); i108143107@nkust.edu.tw (T.-L.N.)

**Abstract:** Logistics services aid import and export businesses located near ports in terms of ease and efficiency in the globalization era. Furthermore, economic growth and global import–export volumes immediately impact the port industry. This research aims to develop a two-stage Data Envelopment Analysis (DEA) model for measuring the performance efficiency of Vietnam's top 18 seaports. The DEA resampling technique is used to forecast future performance, and the DEA Malmquist model analyzes efficiency improvement. First, the forecast data for the next three years, from 2021 to 2023, were obtained by resampling Lucas weight prediction with the highest accuracy. The results indicate that 12 out of all ports achieved an average progressive production efficiency over the entire study period of 2018–2023. Further, most ports have advanced slightly in technological efficiency, indicating that the determinants of increased productivity are the technical efficiency change indexes. This work contributes to the body of knowledge by being the first to apply the resampling technique in conjunction with the Malmquist model to forecast performance efficiency in the domain of the seaport sector. Furthermore, the managerial implications serve as a beneficial reference for operation managers, policymakers, and researchers when comparing the operational efficacy of seaports to diverse logistical scenarios.

**Keywords:** efficiency; forecasting; seaport terminal; DEA; malmquist; resampling; Vietnam

**MSC:** 97M30; 91B02; 62P05; 91B84

## 1. Introduction

The port industry plays a crucial role in the economic prosperity of the nation and region in the age of globalization [1]. Logistics services help import and export businesses located close to ports in terms of ease and efficiency. Seaport terminals are viewed as drivers in the global trading system, acting as a catalyst for the growth of the maritime industry and a significant contribution to a nation's worldwide competitiveness. Port operations are no longer limited to the handling of cargo. They have become a vital component of enterprises and economic markets internationally. Whether for cargo or people, many port facilities are controlled by operators with a global footprint [2]. Terminal operators must overcome unique obstacles and challenges in each nation's seaports to ensure the sector's viability and resilience. As a result, investment in port infrastructure quality enhancement and economic contribution is frequently questioned by authorities, investors, and the general public [3]. In this context, evaluating seaports' operational efficiency, technological capability and progress, and performance productivity is critical for capturing their development trend [4].

In addition, there is a dearth of literature on general supply chains and seaports, especially in developing countries [5]. Vietnam is considered an attractive destination of numerous foreign direct investment projects among emerging countries. It has achieved great strides in international seaborne transport due to the friendly investment environment, low labor costs and reduced taxes compared to India and China [6]. Demand for imported raw materials and completed products between Vietnam and other countries is increasing as the manufacturing landscape changes. Container throughput increased 18% in 2021, reaching over 16.8 million TEUs. Export container throughput was 5.4 million TEUs, up 16% from 2020, while import container throughput was over 5.5 million TEUs, up 21% from 2020 [7].

Moreover, thanks to its favorable geopolitical position with a 1900-mile-long coastline and 320 ports, Vietnam plays an expanding role in global commerce. However, the current development level of Vietnam's seaports is still insufficient compared to the advantages conferred by its geographic location. According to the World Economic Forum, Vietnam has ranked 80/139 nations on the quality of its seaport infrastructure, with an average score of 3.80 on a scale of 1 (lowest) to 7 (highest) between 2006 and 2020. Here, Nam Dinh Vu updates the developing situation of the Vietnam seaport system so that investors can make an informed choice about where to place their business [8]. With a significant number of supply chain relocations to Vietnam, it is natural that shipping demand in the country is expanding. However, given Vietnam's long-term economic trajectory, this need is likely to persist for years to come. Vietnam intends to raise investment funds from various sources, including the state budget and the private sector, to fund its master plan for seaport development between 2021 and 2030 [9]. By 2030, Vietnam's seaports are expected to carry approximately 1.14–1.42 billion tons of cargo, including 38–47 million TEUs of container goods and 10.1–10.3 million passengers annually, according to a recently approved master plan [10]. While this planning process continues, investors that want ports for imports or exports should consider regional variance in the quality of port infrastructure before establishing a facility. Therefore, port efficiency assessment is crucial for port administrators, investors, governments, and clients because it enables an examination of ports' performance and identifies critical factors contributing to port productivity and trade competitiveness [11–13].

Furthermore, performance is a broad concept encompassing nearly all operational management and competitive excellence objectives for a business and its activities. In the case of ports, due to the complexity of today's port products, each actor (authorities, operators, and stakeholders) conducts in-depth examinations of various performance components. These are based on the development of several, occasionally unique performance indicators that differentiate terminal operations, cargo transfer operations, port logistics, and manufacturing and postponed manufacturing activities. The users of these products and the selection criteria for each specific port service vary significantly. With the spatial and functional expansion of ports (port regionalization), connecting and integrating operational design and strategy across the multi-institutional and cross-functional port sector has emerged as a new dimension of port performance. Performance measurements and communication of performance measures that are tailored to specific objectives are becoming more prevalent.

Port-related literature has addressed efficiency from a variety of perspectives. Essentially, port efficiency analyses established relationships between inputs (primarily the physical facilities and labor force of a port) and outputs (such as the quantity or movement of goods in ports). A production or cost frontier—the set of maximum outputs for a given level of inputs or the set of minimum inputs for a given level of outputs—must be estimated. The production frontier, in this context, refers to the optimal combination of inputs in a particular industry. Thus, it is considered inefficient if a producer operates below the frontier. According to this literature, efficiency can be defined as the difference between the position assigned to each observation—which is determined by the relationship between its inputs and outputs—and the estimated best practices on the production frontier [14].

Although port performance is complex and no longer limited to internal procedures, port operators and authorities continue to place a premium on port efficiency [15]. Various support models have been proposed to measure efficiency to solve these concerns over the years. Data envelopment analysis (DEA), proposed by Charnes et al. [16], is known as a popular nonparametric assessment technique based on considering multi-input and multi-output variables as well as decision-making units (DMUs) for efficiency evaluation across geographies or periods [4,17,18]. Various previous studies have applied DEA-based techniques to a variety of domains, including Charnes–Cooper–Rhodes (CCR), Banker–Charnes–Cooper (BCC), super slack-based measure (super SBM), Window DEA, and DEA Malmquist [2,13]. Notably, the CCR and BCC models cannot solve undesirable outputs [19]. The SBM model is used to eliminate slack while accurately ranking the efficacy of DMUs, further optimizing the shortcomings to obtain reasonable metrics. The DEA Malmquist model is an advantageous extension for calculating DMU productivity. The DEA window analysis takes a dynamic approach, treating the same DMU as distinct DMUs at different points in time. The moving average method determines its relative efficiency by comparing each DMU to a different reference set.

Although there are many studies on the DEA measurement of efficiency, to our knowledge, only a few studies have employed the combination of the DEA resampling and Malmquist approach to estimate seaport performance. Therefore, the DEA applications in the literature proved their enormous advantages in efficiency evaluation in the seaport industry. To address this research gap, our study evaluated the efficiency of the top Vietnamese seaports using the combined Malmquist–resampling DEA approach. This study considers the top 18 seaports associated with five major indicators regarding efficiency evaluation in Vietnam, including three input variables: terminal length, equipment, and ship calls. In contrast, cargo throughput twenty-foot equivalent units (TEUs) are chosen as two output variables. The dataset was extracted from Vietnam Seaports Association (VPA) [20] and General Statistics Office (GSO) [21] for the period 2018–2020. Relating these insights to the literature, our main contributions are threefold:

(1) An effective two-stage DEA approach integrating resampling technique and Malmquist is first proposed to assess performance efficiency in the context of the Vietnamese port industry.

(2) The DEA resampling is applied to forecast the next 3 years of seaport performance based on the efficiency score to confirm the suitable data in this case study.

(3) The DEA Malmquist model estimates total productivity change through technical and technological changes based on selected inputs (terminal length, equipment, ship calls) and outputs (cargo throughput, TEUs).

The rest of this study is divided into sections. Section 2 mentions a comprehensive port industry literature review. Section 3 describes the process of the DEA Malmquist and Resampling models. In Section 4, a case study of Vietnam is investigated to demonstrate the procedures' efficacy and the method's applicability to the maritime industry's performance evaluation problem. Section 5 summarizes the study's conclusions, contributions, limitations, and future work.

## 2. Literature Review

In the past decade, many researchers have paid attention to the consideration of production optimization. Debreu [22] provided the first measure of productive efficiency in the context of the coefficient of resource utilization. Farrell [23] presented a similar method for calculating efficiency by considering various inputs and outputs. The term "input-based Farrell efficiency" refers to the highest percentage contraction of all inputs necessary to produce the same amount of output. On the other hand, the output-oriented Farrell efficiency measures the proportionate expansion of all outputs with a given number of inputs. Efficiency can be divided into allocative (cost) and technical. The former denotes the ideal mix of inputs and outputs under the assumption that the producer wishes to minimize costs, whereas the latter refers to the efficiency with which a particular set

of inputs is employed to generate an output. The primary disadvantage of the Farrell efficiency is the weighting of inputs and outputs. CCR [14] is known as a mathematical programming-based optimization method to address this issue. The DEA approach was developed with constant returns to scale or DEA–CCR. This method enables the relative efficiency of DMUs to be determined without the use of fixed weights or time series analysis. DEA–BCC [24] was devised by integrating the BCC model and variable returns to scale (VRS). Since then, the DEA approach has been modified in various ways, including incorporating dummy or categorical variables, discretionary and non-discretionary variables, and nonparametric Malmquist indices. For example, the super SBM model is utilized to address the slack issue while precisely assessing the effectiveness of DMUs [25]. As an expansion of the original DEA model, the DEA Malmquist model is an extremely valuable tool for determining the productivity of DMUs, with the Malmquist productivity index (MPI) being the product of the catch-up index (technical efficiency) and frontier-shift index (technological efficiency) [26]. In this study, the MPI extracted from the Malmquist model is exploited to find the most efficient ports in Vietnam. Due to the suitability of DEA for comparing homogeneity units such as container ports, port benchmarking studies frequently use operational data due to the difficulties of acquiring a port's expenses and pricing [27]. As a result, our study will concentrate on technical efficiency and technological efficiency measurements.

Table 1 summarizes the applications of DEA in the literature to determine the relative efficiency of container terminals and ports. Tongzon [28] evaluated the efficiency of four Australian and twelve other foreign container ports using the CCR and Additive DEA models. Cullinane and Wang [29] utilized CCR and BCC models to analyze the performance of 57 container terminals. Jiang and Li [30] proposed a technique for estimating the technical efficiency of Northeast Asian seaports as a performance metric using DEA radial and non-radial approaches. Sharma and Yu [31] optimized the benchmarks and prioritized the variables in 70 container terminals using a hybrid decision tree in context DEA. Lim et al. [32] used additive non-oriented DEA with RAM, a modified version of the context-dependent DEA model, to assess the relative operating efficiency of 26 Asian container terminals. Sánchez and Millán [33] examined the effect of public changes on the productivity of Spanish ports using the Malmquist index model (MPI). Wanke [34] optimized both steps concurrently using a two-stage network–DEA technique. The first stage, named physical infrastructure efficiency, involves utilizing assets (number of terminals, warehouse space, and yard space) to meet a specified shipping frequency each year. These operations enable the handling of solid bulk and containerized shipments in the second step, named shipment consolidation efficiency. Bray et al. [35] investigated the efficiency of transportation systems using the DEA model based on fuzzy sets. This method was then applied to a sensitivity study of the efficiency of container ports on the Mediterranean Sea involving different attributes. Almawsheki and Shah [36] measured technical efficiency using the CCR model, and slack variable analysis identified possible areas for improvement at 19 inefficient container terminals in the Middle Eastern region. Sun et al. [37] suggested a non-radial DEA preference model based on the VRS and the directional distance function (DDF) for analyzing the efficiency of Chinese-listed port firms. Huang et al. [38] employed the DEA–supply chain operations reference (SCOR) model to examine the port operational efficiency of ship inward–outward and stacking yards in ports along the Maritime Silk Road of the twenty-first century. Mustafa et al. [39] applied DEA–CCR and DEA–BCC models to compare the technical efficiency of less-explored South Asian and Middle Eastern ports to East Asian ports and identify opportunities for improving their efficiency and management. Based on the idea of supply base rationalization in Korea container terminals, Kim et al. [40] suggested DEA cross-efficiency and cluster analysis to assess the direction for terminal rationalization at the national level in order to alleviate excessive rivalry among container terminals. Xu and Xu [41] integrated the exponential smoothing method with the DEA model to determine the business plan for SIPG during the next five years. Liu et al. [42]

used the SBM–DEA and undesirable–DEA models to assess the efficiency of the major container terminals in three Chinese cities.

**Table 1.** A summary of previous study methodologies and problem features.

| Authors/Year | Inputs/Criteria | Outputs/Responses | Method | Sample and Region |
|---|---|---|---|---|
| Tongzon (2001) [28] | Number of cranes Number of container berths Number of tugs Terminal area Delay time labor | Throughput Number of ship calls | CCR Additive DEA. | Brazilian ports |
| Cullinane and Wang (2006) [29] | Terminal area Quay cranes Yard cranes straddle carriers | Container throughput | CCR BCC | container ports |
| Jiang and Li (2009) [30] | Import/Export by customs GDP by regions Berth length Crane number | Container throughput | Radial Non-radial | Northeast Asian container ports |
| Sharma and Yu (2010) [31] | Quay cranes Transfer cranes Straddle carriers Reach stackers Quay length terminal area | Container throughput | Context-DEA | Container terminals |
| Lim et al. (2011) [32] | Quay length Total area Gantry cranes | Container throughput | Additive non-oriented DEA RAM | Asian container terminals |
| Sánchez and Millán (2012) [33] | Number of employees Intermediate consumption Capital | Liquid bulk solid bulk Containerized general cargo Non-containerized general cargo | MPI | Ports in Spain |
| Wanke (2013) [34] | Number of berths Warehousing area yard area Shipments frequency | Container throughput | Network-DEA | Brazilian ports |
| Bray et al. (2014) [35] | Number of cranes Container berths Number of tugs Terminal area Delay time Number of port authority employees | Container throughput Shiprate Ship calls Crane Productivity | Fuzzy DEA | Container ports |
| Almawsheki and Shah (2015) [36] | Terminal Area Quay length Quay cranes Yard equipment Maximum Draft | Container throughput | CCR | Middle East container terminals |
| Sun et al. (2017) [37] | Staff number Fixed assets | Operating cost Net profit Cargo throughput NOx | Non-radial DEA | Chinese port enterprises |

**Table 1.** *Cont.*

| Authors/Year | Inputs/Criteria | Outputs/Responses | Method | Sample and Region |
|---|---|---|---|---|
| Huang et al. (2021) [38] | Quay length Number of container berths Gantry cranes | Container throughput | CCR BCC SCOR | Ports along the twenty-first-century Maritime Silk Road |
| Mustafa et al. (2021) [39] | Number of berths Number of cranes Berth length Berth depth | TEUs | CCR BCC | Ports in South & Middle Eastern and East Asian region |
| Kim et al. (2021) [40] | Quay length Depth of water Crane | Cargo volume Loading capacity per hour | DEA cross-efficiency Cluster analysis | Korean Container Terminals |
| Xu and Xu (2021) [41] | R&D Proportion of technical personnel | Business income Container throughput | Exponential smoothing CCR | Ports in China |
| Liu et al. (2022) [42] | Gross Crane Productivity Crane Intensity Berth Length Berth Depth | Calls Moves Finish | SBM Undesirable | Ports in China |

Based on the extensive literature review, a few prior studies forecasted and evaluated the port industry's performance using a combined DEA approach of Resampling forecasting techniques and the Malmquist model. This paper, therefore, is not concerned with the description of critical factors influencing seaport efficiency in general, but rather with identifying gaps in the scope of previous studies on seaport management and beginning to address those gaps through an empirical study of the top 18 seaports in Vietnam. The goal is to increase knowledge and provide a better understanding of stakeholder management issues in the context of the Vietnamese port industry.

According to Chang et al. [43], it is critical to consider the DMU's past, current performance, and future potential when evaluating its performance. However, if the past, current, and future performance are all considered concurrently, it is necessary to integrate various methodologies. Tone [44] suggested a model in DEA to address this issue, called resampling past–present and resampling past–present–future. The past–present model estimates the DEA score's confidence interval over the past and present time periods using the super slack-base measure model (super-SBM), and then extends this model to the past–present–future time periods. Wang et al. [45] used this DEA resampling model to forecast the macroeconomic performance of 17 economies, including 12 Asian developing countries and five developed countries, during 2013–2020. The research demonstrates that DEA resampling is a highly successful model for forecasting and evaluating the performance of numerous decision makers. Chiu et al. [46] developed a method for evaluating the financial industry's performance in Taiwan by combining the merger potential gains model and the resample past–present–future model. Bai et al. [47] used a combination of resampling DEA and the possible merger benefits model to estimate the efficiency gains associated with three representative mergers and acquisitions schemes in China's railway sector from 2011 to 2015.

The Malmquist model is an extremely useful tool for evaluating productivity in DEA [48]. Färe [49] indicated that the Malmquist model consists of two components, one of which assesses changes in technical efficiency and the other of which measures changes in technology efficiency. The Malmquist model measures a DMU's total factor productivity change over a two-year period. It is defined as the product of efficiency improvement (catch-up) and technological advancement (frontier-shift). The catch-up effect describes how near a DMU gets to the most efficient production frontier and the frontier-shift effect describing the sample's technological advancement. The deconstructed

aspects of the MPI can assess how much of an increase in relative efficiency from period t to can be attributed to individual effort and how much to industry development. Efficiency change quantifies the degree to which a DMU improves or decreases its efficiency, whereas technological change measures the efficiency of frontier shift between two periods [50]. There have been various applications of the Malmquist model over time in various fields. For example, Pan et al. [51] used the Malmquist model to examine regional disparities and the dynamic evolution of agricultural sustainability and efficiency in 31 regions of Mainland China. Wan and Zhou [52] used the Malmquist model to measure the total productivity of agricultural management factors of 12 cities in Hubei, a central province of China. Kong et al. [53] suggested that the Malmquist model investigates the development efficiency, spatiotemporal evolution characteristics, and spatial improvement of China's innovative industrial clusters.

Therefore, this study uses a hybrid technique for forecasting and evaluating port industry performance by a combined DEA approach of Resampling and Malmquist models. First, Resampling is used to forecast future values for each seaport for the period 2021–2023, and Malmquist is applied to determine the efficiency changes score over the entire period of 2018–2023 based on output variables such as cargo throughput and TEUs, and input variables such as terminal length, equipment, and ship calls. This study serves as a guideline for managers, policymakers, and decision makers to optimize operative processes and identify critical success criteria for sustainable growth, harnessing and enhancing Vietnam's port industry.

## 3. Methodology

The present study employs an integrated DEA Malmquist and resampling model to evaluate Vietnam's top 18 seaports for 2018–2020. After thoroughly examining the Vietnamese port industry, this study carefully considers the top 18 seaports. Selecting appropriate input and output variables is critical in DEA applications, as the correlation between the selected variables will affect the accuracy of the results. The research process is divided into two distinct phases, as illustrated in Figure 1.

First, Pearson correlation is checked to ensure the dataset's homogeneity and isotonicity. Second, the DEA resampling technique forecasts the performance efficiency of 18 selected ports using historical data from 2018 to 2020. The authors then propose resampling the past–present–future model to analyze future results from 2021 to 2023. Finally, the DEA Malmquist model determines the total productivity change. The researchers present the results of their application of the Malmquist productivity index, analyze the data over time, and summarize the findings.

### 3.1. Validation of Data

Numerous statistical techniques such as Pearson correlation coefficient, affinity index, diversity index, etc. are used to validate data. The correlation between the input and output data will be verified before calculating the efficiency. Pearson's correlation coefficient test is a well-known method that has been utilized in past investigations. Scores are composed of values ranging from $-1$ to $+1$ in relation to each score. Each score reflects the linear dependence between two determinants or data sets [54]. The homogeneity and isotonicity will demonstrate that correlation tests are significant, allowing for any DEA methodologies. Getting correlation values close to $+1$ indicates a more favorable linear relationship in simple terms.

$$r_{xy} = \frac{\sum_{i=1}^{n}(x_i - \overline{x})(y_i - \overline{y})}{\sqrt{\sum_{i=1}^{n}(x_i - \overline{x})^2}\sqrt{\sum_{i=1}^{n}(y_i - \overline{y})^2}} \tag{1}$$

where $n$ denotes the sample size, $x_i$, $y_i$ are the individual points indexed $i$, and $\overline{x} = \frac{1}{n}\sum_{i=1}^{n} x_i$ is the sample mean and analogous for $\overline{y}$.

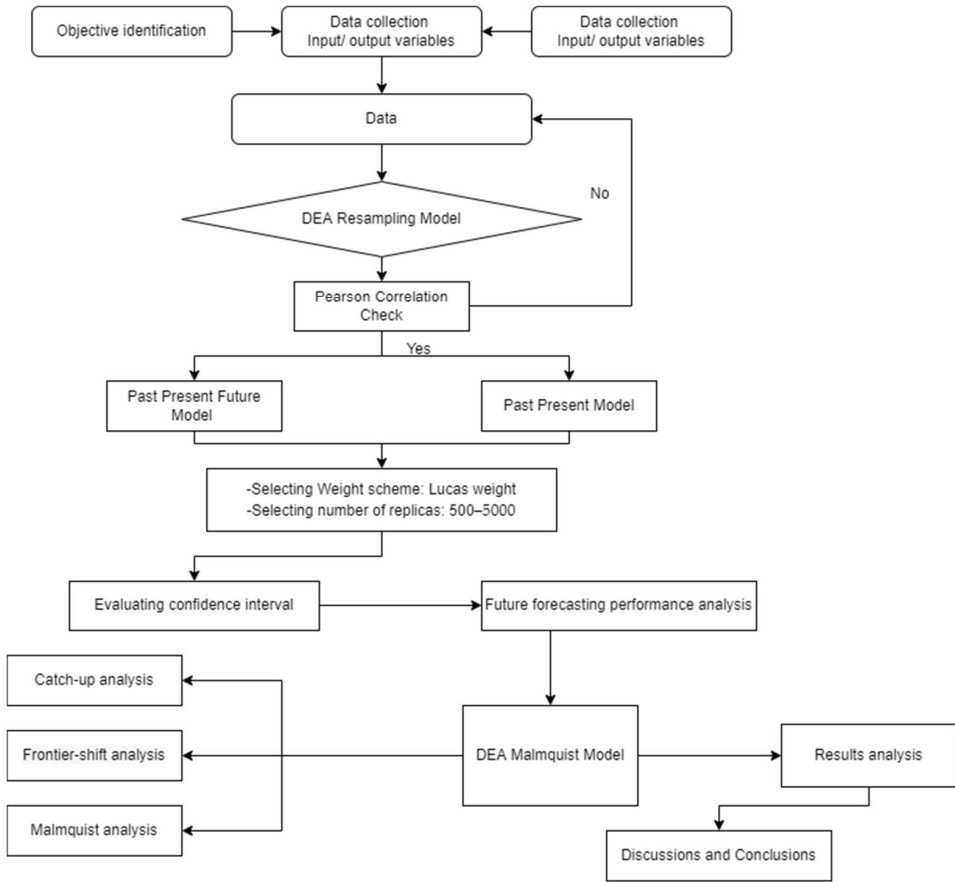

**Figure 1.** Research framework.

### 3.2. DEA Resampling Model

Since its establishment, DEA has been continuously enhanced, and appropriate methodologies have been developed. While these approaches inherit two desirable qualities, unit invariance and monotonicity, they share a similar flaw in that they are measured only once, resulting in some computation bias [47]. To address this shortcoming, it is necessary to consider their history, current record, and future potential. Incorporation of multiple approaches, such as DEA and other forecasting models such as time series models [55], Grey models [56], and machine learning forecasting algorithms [57], is required when considering the past, current record, and future production.

Tone [58] suggested three DEA resampling models. The resampling model is primarily concerned with input and output measurement errors and how repeated sampling has reduced some empirical distribution flaws. Additionally, this method may forecast the future efficiency of DMU and increase forecast accuracy more than GM (1,1).

#### 3.2.1. Past-Present Model

$(X^t, Y^t)$ $(t = 1, \ldots, m)$ is denoted as a set of historical data for resampling, with the input vector $X^t = (x_1^t, \ldots, x_n^t)$ $(x_j^t \in R^m)$ and the output vector $Y^t = (y_1^t, \ldots, y_n^t)$ $(y_j^t \in R^g)$, with n DMUs. $W_t$ is denoted for a time weight increasing with $t$.

Lucas number series $(Wl_1, \ldots, Wl_t)$, is defined as follows:
$Wl_{t+2} = Wl_t + Wl_{t+1}$ $(t = 1, \ldots, m, m - 2; l_1 = 1, l_2 = 2)$.

$WL$ stands for the sum of the series: $WL = \sum\limits_{t=1}^{m} Wl_t$. Then, the weight $W_t$ is as follows:

$$W_t = Wl_t / WL \tag{2}$$

Finally, a bootstrap replication of historical data is used to calculate the confidence interval. Because replicas are representative of the dataset, it is necessary to conduct a preliminary analysis of the data to determine its characteristics. The 95 percent confidence interval can then be calculated using Fisher's transformation [19].

### 3.2.2. Past–Present–Future Model

$\left(X^{t+1}, Y^{t+1}\right)$ is denoted as the forecast data, calculated by taking past-present data $\left(X^t, Y^t\right)$ with $(t = 1, \ldots, m)$ and measuring the DMU efficiency value in the future period alongside their confidence intervals.

$F^t(t = 1, \ldots, m)$ is denoted as the historical data with an exact input and output of a DMU. $Fh^{m+1}$ is forecasted from $F^t(t = 1, \ldots, m)$. There are three resampling techniques: trend analysis, Lucas weight average, and a hybrid of trend and Lucas.

### 3.3. DEA Malmquist Model

The Malmquist Index was suggested by Caves et al. [48] to compare the technological efficiency of a range of items. The Malmquist Index quantifies a DMU's overall factor productivity change over two consecutive periods. It is characterized by efficiency changes (catch-up) and technological changes (frontier–shift). A change in efficiency indicates how much the DMU improves or degrades its efficacy, whereas technical changes indicate a shift in the borders of efficiency between two periods [59].

The MPI index = Efficiency change (catch-up) × Technical change (frontier–shift):

$$\text{MPI}\left(x^{t+1}, y^{t+1}, x^t, y^t\right) = \frac{\rho_0^{t+1}\left(x^{t+1}, y^{t+1}\right)}{\rho_0^t\left(x^t, y^t\right)} \sqrt{\frac{\rho_0^t\left(x^{t+1}, y^{t+1}\right)}{\rho_0^{t+1}\left(x^t, y^t\right)} \times \frac{\rho_0^t\left(x^t, y^t\right)}{\rho_0^{t+1}\left(x^t, y^t\right)}} \tag{3}$$

It is well noted that MPI values can be extracted in three scenarios:

- MPI values > 1: Increasing productivity
- MPI values = 1: Constant productivity
- MPI values < 1: Decreasing productivity.

## 4. Empirical Analysis

### 4.1. Case Study

This study investigates the top 18 seaports in Vietnam from 2018–to 2020 via DEA-Solver software Version 13.2 (Table 2).

**Table 2.** List of DMUs.

| DMUs | Seaport | Area |
|------|---------|------|
| SP-01 | Quang Ninh | Northern |
| SP-02 | Hai Phong | Northern |
| SP-03 | Doan Xa | Northern |
| SP-04 | Dinh Vu | Northern |
| SP-05 | Nam Dinh Vu | Northern |
| SP-06 | Tan Cang 128 | Northern |
| SP-07 | Nghe Tinh | Central |
| SP-08 | Da Nang | Central |
| SP-09 | Quy Nhon | Central |
| SP-10 | Dong Nai | Southern |
| SP-11 | Cat Lai | Southern |
| SP-12 | Sai Gon | Southern |
| SP-13 | Ben Nghe | Southern |
| SP-14 | Lotus | Southern |
| SP-15 | TCIT+TCCT | Southern |
| SP-16 | SSIT | Southern |
| SP-17 | Can tho | Southern |
| SP-18 | An Giang | Southern |

The accuracy of the data is critical because it has the potential to alter the results significantly. Based on the summary of inputs and outputs used in previous relevant studies, the authors decided to choose three inputs (Terminal Length, Equipment, Ship calls) and two outputs (Cargo throughput, TEUs), which are presented in Table 3 [13].

**Table 3.** Definitions of variables.

| Variables | Definitions | Units | References |
|---|---|---|---|
| Terminal Length ($I_1$) | The length of berths at which container ship anchor | $m^2$ | [2,13] |
| Equipment ($I_2$) | The major number of equipment cargo-handling in port | Items | [13,60] |
| Ship calls ($I_3$) | The number of vessels which call or arrive at a particular port at any given time | Call | [11,12] |
| Cargo throughput ($O_1$) | The weighted quantity of cargo handled annually | MT | [15,61] |
| TEUs ($O_2$) | The terminal's annual container | TEU | [13,15] |

*4.2. Results of DEA Resampling Model*

4.2.1. Results of DEA Resampling Model for Historical Data

This study uses historical data of the top 18 seaports for three years (2018–2020) to calculate the efficiency scores using Lucas weights. A replicas test evaluates correlation coefficients for the study's findings. The comparisons of 5000 versus 500 replicates are provided in Table 4, indicating that the results obtained with a 95% confidence interval were statistical negligibly low, while the DEA scores are virtually identical. Thus, 500 replicas can be used in this study.

Before further analysis, a correlation analysis is performed to ensure that the appropriate inputs and outputs are used in the DEA models. The correlation coefficient value is always between ($-1$) and ($+1$), and if it is close to (1), it indicates a stronger linear relationship between the components. The high correlation between inputs and outputs confirms that the study's inputs and outputs are appropriate, as indicated by the correlation analysis results presented in Table 5.

**Table 4.** Comparisons of 5000 and 500 replicas (2018–2020).

| DMUs | 5000 Replicas | | | 500 Replicas | | | Difference | |
|---|---|---|---|---|---|---|---|---|
| | 97.50% | DEA | 2.50% | 97.50% | DEA | 2.50% | 97.50% | 2.50% |
| SP-01 | 1.0107 | 0.7801 | 0.6251 | 1.0082 | 0.7801 | 0.6251 | 0.0025 | 0 |
| SP-02 | 1.1436 | 0.8952 | 0.6545 | 1.1449 | 0.8952 | 0.6623 | −0.0013 | −0.0078 |
| SP-03 | 0.3698 | 0.2484 | 0.1077 | 0.3698 | 0.2484 | 0.1083 | 0 | −0.0006 |
| SP-04 | 2.0137 | 1.3668 | 1.3012 | 2.006 | 1.3668 | 1.2965 | 0.0077 | 0.0047 |
| SP-05 | 1.4568 | 1.0705 | 0.6937 | 1.4653 | 1.0705 | 0.6834 | −0.0085 | 0.0103 |
| SP-06 | 1.3138 | 0.376 | 0.3236 | 1.3079 | 0.376 | 0.3236 | 0.0059 | 0 |
| SP-07 | 0.2325 | 0.1966 | 0.1805 | 0.2385 | 0.1966 | 0.1791 | −0.006 | 0.0014 |
| SP-08 | 0.4094 | 0.3855 | 0.2377 | 0.4174 | 0.3855 | 0.2373 | −0.008 | 0.0004 |
| SP-09 | 0.5723 | 0.5266 | 0.3683 | 0.5763 | 0.5266 | 0.3872 | −0.004 | −0.0189 |
| SP-10 | 2.909 | 1.1525 | 0.5062 | 2.9088 | 1.1525 | 0.5007 | 0.0002 | 0.0055 |
| SP-11 | 0.4002 | 0.2801 | 0.263 | 0.4229 | 0.2801 | 0.263 | −0.0227 | 0 |
| SP-12 | 0.3883 | 0.3367 | 0.2613 | 0.3918 | 0.3367 | 0.2613 | −0.0035 | 0 |
| SP-13 | 0.5701 | 0.4683 | 0.3766 | 0.5735 | 0.4683 | 0.3766 | −0.0034 | 0 |
| SP-14 | 0.1772 | 0.1279 | 0.0519 | 0.1787 | 0.1279 | 0.0536 | −0.0015 | −0.0017 |
| SP-15 | 5.9669 | 2.3891 | 3.9871 | 6.0392 | 2.3891 | 3.9797 | −0.0723 | 0.0074 |
| SP-16 | 1.2658 | 1.1213 | 0.2484 | 1.2665 | 1.1213 | 0.2585 | −0.0007 | −0.0101 |
| SP-17 | 0.2038 | 0.1771 | 0.1088 | 0.2054 | 0.1771 | 0.1088 | −0.0016 | 0 |
| SP-18 | 0.4413 | 0.4098 | 0.3703 | 0.4441 | 0.4098 | 0.3689 | −0.0028 | 0.0014 |

**Table 5.** Correlation matrix of inputs and outputs in 2020.

|  | Terminal Length | Equipment | Ship Calls | Cargo Throughput | TEUs |
|---|---|---|---|---|---|
| **Terminal Length** | 1.000 | 0.981 | 0.852 | 0.912 | 0.922 |
| **Equipment** | 0.981 | 1.000 | 0.829 | 0.953 | 0.968 |
| **Ship calls** | 0.852 | 0.829 | 1.000 | 0.834 | 0.795 |
| **Cargo throughput** | 0.912 | 0.953 | 0.834 | 1.000 | 0.988 |
| **TEUs** | 0.922 | 0.968 | 0.795 | 0.988 | 1.000 |

4.2.2. Results of DEA Resampling Model for Future Data

To forecast future performance, the authors consider the period between 2018 and 2020 to be past–present, while the period between 2021 and 2023 is considered future. Second, three distinct projections are used to forecast the year 2020's efficiency (trend, Lucas weight, and a hybrid model combining the trend and Lucas weight models). The actual efficiency score obtained with the Super-SBM model will then be compared to the forecast score in order to determine the prediction model's accuracy. As previously stated, 500 replicas are acceptable; thus, this section forecasted future operations of 18 ports in Vietnam using 500 replicas with a 95% confidence interval. After calculating and comparing the actual score for 2020 with the efficiency scores obtained through three projections, it was determined that the actual scores for all 18 sample ports included in the 95% confidence interval and the average Forecast-Actual by Lucas weight prediction are 21,06%, which is the lowest of the three separate projections. As a result, Lucas weight prediction is used to forecast data for the three years 2021–2023. Table 6 compares the actual and forecast score (as predicted by the Lucas weight model) for 2020.

**Table 6.** Forecast scores by the Lucas weight model, actual scores, and confidence interval in 2020.

| DMUs | 97.50% | Forecasted Score | Actual Score | 2.50% |
|---|---|---|---|---|
| SP-01 | 1.0346 | 0.9746 | 0.7801 | 0.6427 |
| SP-02 | 1.1402 | 1.0625 | 0.8952 | 0.6322 |
| SP-03 | 0.3687 | 0.1907 | 0.2484 | 0.1100 |
| SP-04 | 2.1020 | 1.4263 | 1.3794 | 1.3219 |
| SP-05 | 1.4928 | 1.1642 | 1.1065 | 0.7053 |
| SP-06 | 1.2317 | 0.5695 | 0.3760 | 0.4479 |
| SP-07 | 0.2386 | 0.2187 | 0.1966 | 0.1978 |
| SP-08 | 0.3011 | 0.2566 | 0.3855 | 0.2242 |
| SP-09 | 0.4601 | 0.4251 | 0.5266 | 0.3718 |
| SP-10 | 2.7488 | 1.9320 | 1.7165 | 0.4815 |
| SP-11 | 0.4002 | 0.3369 | 0.2801 | 0.3074 |
| SP-12 | 0.3879 | 0.3639 | 0.3367 | 0.2799 |
| SP-13 | 0.5719 | 0.5321 | 0.4683 | 0.4064 |
| SP-14 | 0.1702 | 0.0912 | 0.1279 | 0.0521 |
| SP-15 | 5.6958 | 4.9512 | 4.5446 | 3.9764 |
| SP-16 | 0.3943 | 0.2900 | 1.1352 | 0.2198 |
| SP-17 | 0.1859 | 0.1395 | 0.1771 | 0.1116 |
| SP-18 | 0.4373 | 0.4086 | 0.4098 | 0.3759 |

*4.3. Results of DEA Malmquist Model*

4.3.1. Technical Efficiency Change

The catch-up index displayed in Table 7 and Figure 2 reflects the evolution of the DMUs' technological efficiency during 2018–2023. The evolution of the catch-up indexes for all ports is depicted in Figure 2, and the detailed catch-up values are reported in Table 4. The catch-up index indicates the advancement and regress of the DMUs' technical efficiency with values greater than or equal to one. Overall, the average catch-up score for all DMUs fluctuated dramatically from 2018 to 2020, and is expected to grow or decrease over 2020–2023. Table 7, 13 ports in all DMUs showed progressive technical efficiency

between 2018 and 2023 with average catch-up indexes >1, resulting in an average catch-up of 1.0518. Among these, SP-16 (1.373), SP-12 (1.3641), and SP-10 (1.209) are the three DMUs with the greatest average gain in technical efficiency between 2018 and 2023. Meanwhile, SP-01 (0.977), SP-04 (0.946), SP-06 (0.9128), SP-11 (0.9803), SP-12 (0.9976), and SP-13 are the least effective operations on average. As illustrated in Figure 2, SP-03, SP-08, and port 16 exhibited the most volatile performance during the research period. In particular, SP-03 and SP-16 achieved their technical efficiency peaks during the 2019–2020 period, with catch-up indexes of 2.2371 and 3.2211, respectively. Furthermore, port-16 is predicted to experience a decline in technical efficiency between 2020 and 2021, with a catch-up index of 0.4971, making it the worst-performing operator, followed by an upward trend in subsequent periods in a score of 0. 9938 at the end of the forecast period. SP-06 raised its catch-up score from 0.4777 in 2018–2019 to 0.7826 in 2019–2020 and is anticipated to continue increasing significantly over the next three years, but has by far the worst catch-up index of the entire year period. The technical efficiency progressions of the remaining DMUs indicate relatively stable patterns.

**Table 7.** Technical efficiency changes for the period 2018–2023.

| Frontier | 2018–2019 | 2019–2020 | 2020–2021 | 2021–2022 | 2022–2023 | Average |
|---|---|---|---|---|---|---|
| SP-01 | 0.8013 | 0.9098 | 1.1547 | 1.0192 | 1.0017 | 0.9773 |
| SP-02 | 1.0079 | 0.8499 | 1.1825 | 0.9985 | 0.9999 | 1.0077 |
| SP-03 | 0.3028 | 2.2371 | 0.8811 | 1.0271 | 1.0024 | 1.0901 |
| SP-04 | 0.6178 | 1.0791 | 1.0180 | 1.0153 | 1.0014 | 0.9463 |
| SP-05 | 1.1625 | 0.9076 | 1.0195 | 0.9939 | 0.9995 | 1.0166 |
| SP-06 | 0.4777 | 0.7826 | 1.2653 | 1.0354 | 1.0030 | 0.9128 |
| SP-07 | 1.1312 | 0.8632 | 1.0488 | 1.0013 | 1.0001 | 1.0089 |
| SP-08 | 0.8359 | 1.5990 | 0.8326 | 0.9838 | 0.9985 | 1.0500 |
| SP-09 | 1.1683 | 1.1768 | 0.9042 | 0.9836 | 0.9985 | 1.0463 |
| SP-10 | 1.9953 | 1.0825 | 0.9671 | 0.9985 | 0.9999 | 1.2087 |
| SP-11 | 0.9610 | 0.8390 | 1.0888 | 1.0115 | 1.0010 | 0.9803 |
| SP-12 | 0.9343 | 1.0099 | 1.0408 | 1.0029 | 1.0003 | 0.9976 |
| SP-13 | 0.9181 | 0.9566 | 1.0690 | 1.0067 | 1.0006 | 0.9902 |
| SP-14 | 1.8819 | 1.0703 | 0.9076 | 0.9574 | 0.9924 | 1.1619 |
| SP-15 | 1.2128 | 1.1298 | 0.9104 | 0.9839 | 0.9986 | 1.0471 |
| SP-16 | 1.2213 | 3.2211 | 0.4971 | 0.9320 | 0.9938 | 1.3731 |
| SP-17 | 1.8041 | 0.9582 | 0.8462 | 0.9617 | 0.9968 | 1.1134 |
| SP-18 | 1.0297 | 0.9915 | 0.9986 | 0.9988 | 0.9999 | 1.0037 |
| Average | 1.0813 | 1.2036 | 0.9796 | 0.9951 | 0.9993 | 1.0127 |
| Max | 1.9953 | 3.2211 | 1.2653 | 1.0354 | 1.003 | 1.0839 |
| Min | 0.3028 | 0.7826 | 0.4971 | 0.932 | 0.9924 | 0.951 |
| SD | 0.4503 | 0.6077 | 0.1686 | 0.0255 | 0.0027 | 0.0372 |

4.3.2. Technological Efficiency Change

The frontier-shift index measures the technological improvements (efficiency frontiers) of DMUs during the period 2018–2023, reflecting their performance under a few variables such as competitiveness, technological change, development, and political and regulatory environment, to highlight a few. Table 8 summarizes the detailed frontier-shift values for the DMUs, and Figure 3 illustrates the evolutionary trajectories of technical efficiencies for all DMUs. Overall, the average frontier-shift indexes of all DMUs fluctuated modestly in 2016–2019 but are expected to increase at a relatively stable rate during the forecast period 2020–2023. Table 8 shows that seven the DMUs, including SP-03 (0.9989), SP-07 (0.9995), SP-08 (0.9963), SP-09 (0.9970), SP-16 (0.9920), SP-17 (0.9949), and SP-18 (0.9943) failed to meet the advancing average frontier-shift indexes. Meanwhile, port-04 (1.0370), SP-06 (1.0446), and SP-11 (1.0567) are the best technologically efficient operators. Due to the fact that the majority of DMUs (eleven out of fourteen) have positive average frontier-shift indexes, the overall average frontier-shift score during the research period is 1.0129 (efficiency increase). In comparison to Figure 1 (catch-up index), Figure 3

shows more fluctuating patterns of DMUs, notably SP-02, SP-04, SP-05, SP-06, SP-11, SP-14, and SP-15, which depict the progression of their technological performance. Among these, SP-02, SP 14, and SP-16 experienced a remarkable increase in 2019–2020 to peaks of 1.1827, 1.1192, and 1.0299, respectively, but are predicted to decline significantly in 2020–2021, followed by a slight increase in the next forecast period, while other DMUs tend to grow in the opposite direction. Especially, 3 DMUs are predicted to be unchanged in 2022–2023 with a score index of 1; this proves that all DMUs are predicted to have no technological change progression.

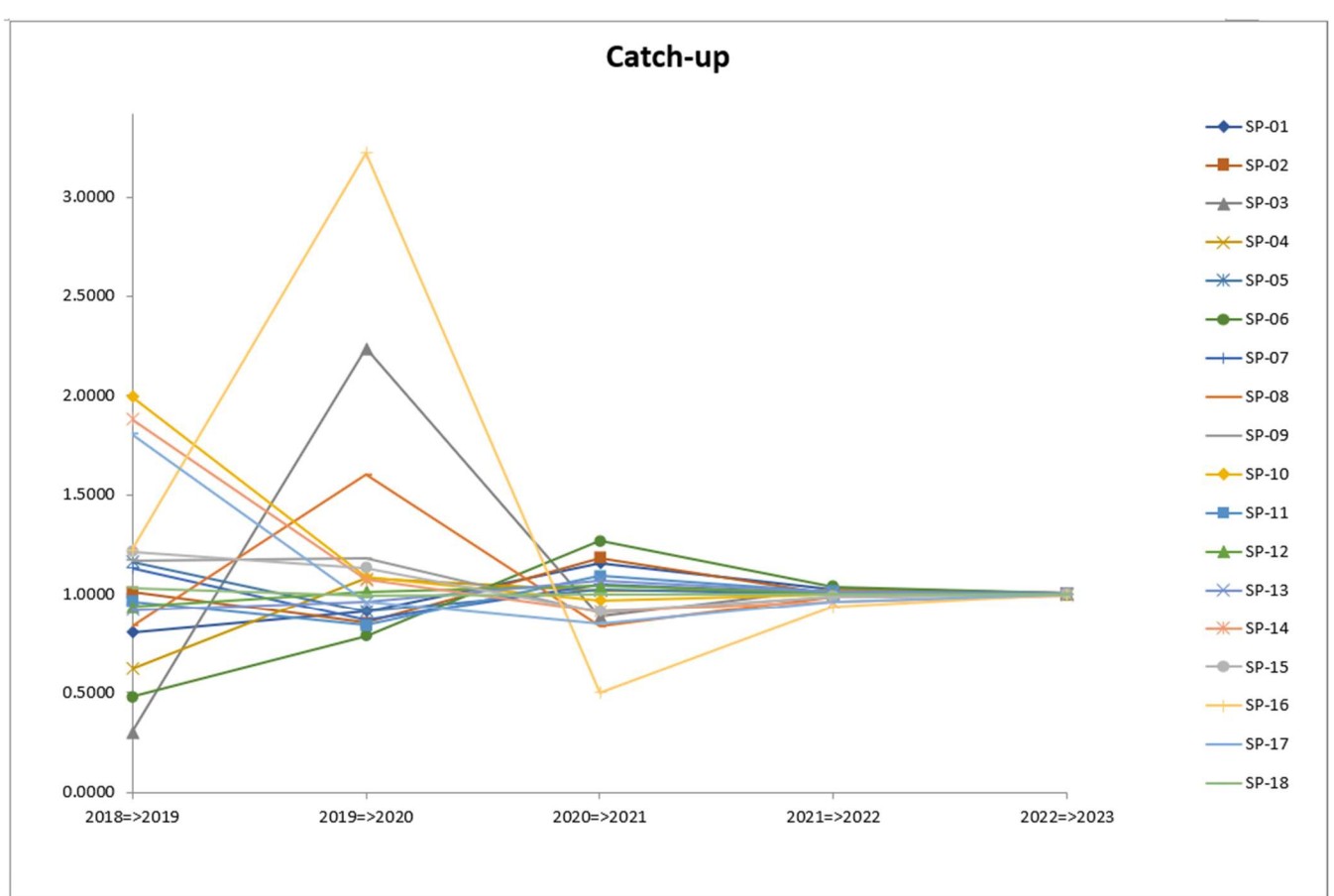

**Figure 2.** Technical efficiency change (catch-up).

**Table 8.** Technological change for the period 2018–2023.

| Frontier | 2018–2019 | 2019–2020 | 2020–2021 | 2021–2022 | 2022–2023 | Average |
|---|---|---|---|---|---|---|
| SP-01 | 1.0424 | 1.0992 | 0.8886 | 0.9899 | 0.9991 | 1.0038 |
| SP-02 | 1.1074 | 1.1827 | 0.8435 | 0.9938 | 0.9994 | 1.0254 |
| SP-03 | 0.9787 | 1.0265 | 0.9895 | 0.9997 | 1.0000 | 0.9989 |
| SP-04 | 1.2125 | 0.9858 | 0.9929 | 0.9943 | 0.9995 | 1.0370 |
| SP-05 | 1.1789 | 1.0136 | 0.9862 | 0.9907 | 0.9992 | 1.0337 |
| SP-06 | 1.2662 | 0.9684 | 0.9887 | 0.9996 | 1.0000 | 1.0446 |
| SP-07 | 0.9788 | 1.0259 | 0.9927 | 1.0001 | 1.0000 | 0.9995 |
| SP-08 | 0.9629 | 1.0226 | 0.9950 | 1.0009 | 1.0001 | 0.9963 |
| SP-09 | 0.9660 | 1.0241 | 0.9940 | 1.0006 | 1.0001 | 0.9970 |
| SP-10 | 1.0644 | 1.0732 | 0.9074 | 0.9792 | 0.9981 | 1.0044 |
| SP-11 | 1.2625 | 1.1304 | 0.9099 | 0.9822 | 0.9984 | 1.0567 |
| SP-12 | 1.0848 | 0.9291 | 0.9889 | 0.9995 | 1.0000 | 1.0005 |
| SP-13 | 1.0884 | 0.9459 | 0.9838 | 0.9983 | 0.9999 | 1.0033 |

**Table 8.** *Cont.*

| Frontier | 2018–2019 | 2019–2020 | 2020–2021 | 2021–2022 | 2022–2023 | Average |
|----------|-----------|-----------|-----------|-----------|-----------|---------|
| SP-14 | 1.0859 | 1.1192 | 0.9190 | 0.9918 | 0.9999 | 1.0232 |
| SP-15 | 1.1362 | 1.0593 | 0.9531 | 0.9905 | 0.9992 | 1.0276 |
| SP-16 | 0.9484 | 1.0299 | 0.9832 | 0.9987 | 0.9999 | 0.9920 |
| SP-17 | 0.9577 | 1.0267 | 0.9897 | 1.0006 | 1.0001 | 0.9949 |
| SP-18 | 0.9514 | 1.0201 | 0.9983 | 1.0016 | 1.0001 | 0.9943 |
| Average | 1.0707 | 1.0379 | 0.9614 | 0.9951 | 0.9996 | 1.0129 |
| Max | 1.2662 | 1.1827 | 0.9983 | 1.0016 | 1.0001 | 1.0567 |
| Min | 0.9484 | 0.9291 | 0.8435 | 0.9792 | 0.9981 | 0.9920 |
| SD | 0.1072 | 0.0647 | 0.0466 | 0.0066 | 0.0006 | 0.0200 |

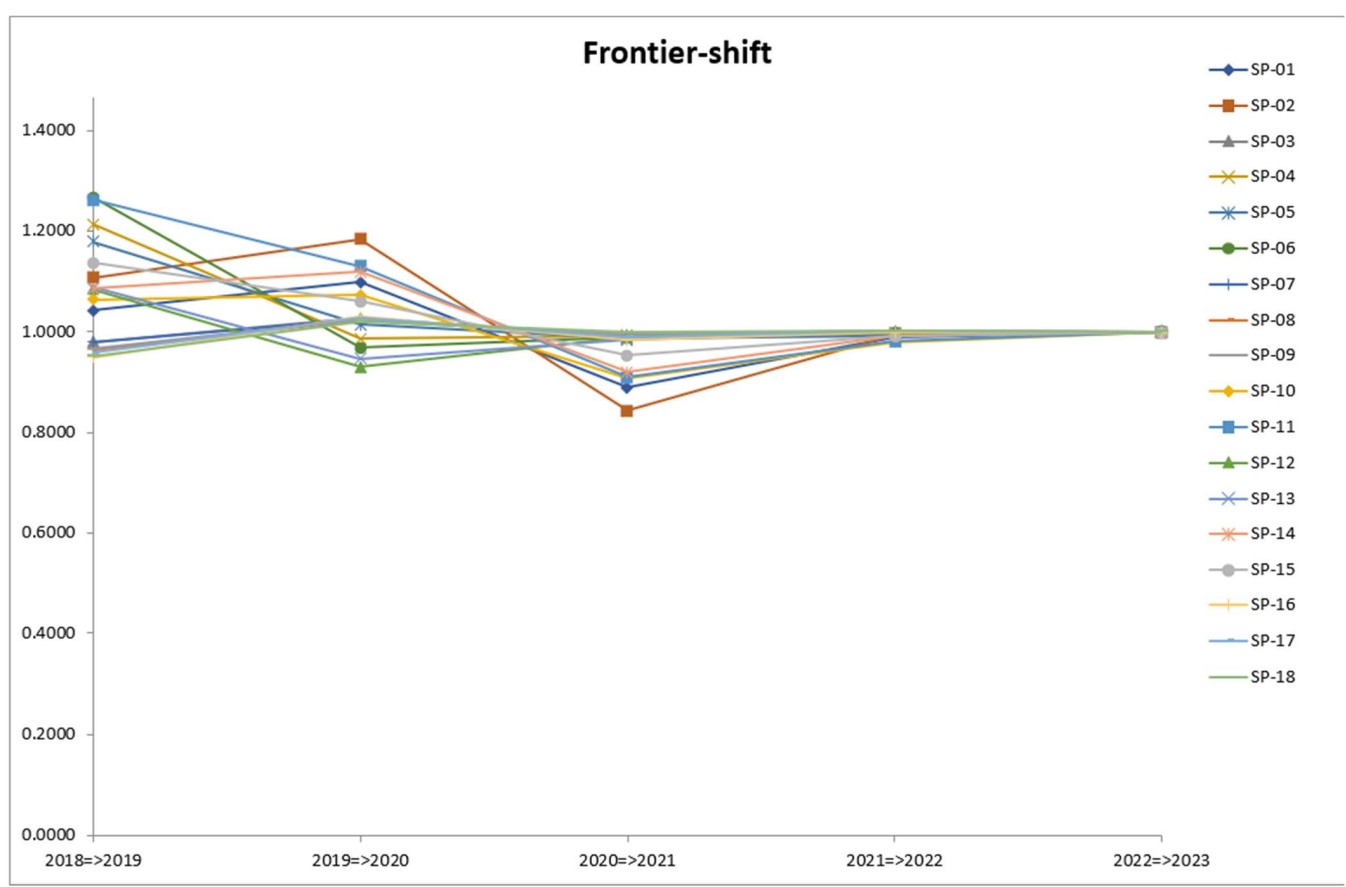

**Figure 3.** Technological change (frontier-shift).

### 4.3.3. Total Productivity Change

The Malmquist Productivity Indexes (MPIs) of the DMUs are obtained using Equation (3). Table 9 contains the detailed MPI indexes, whereas Figure 4 illustrates the progression of the MPIs for all ports. MPI = 1 denotes the status quo for constant efficiency, MPI > 1 denotes an efficiency improvement, and MPI < 1 denotes an efficiency drop. As seen in Table 6, most DMUs performed efficiently on average, except SP-01, SP-04, SP-06, SP-12, SP-13, and SP-18. This result is noteworthy because these DUMs performed poorly with technical efficiency, while there is no noticeable improvement in technological efficiency, as demonstrated by their catch-up and frontier-shift scores discussed in 4.3.1 and 4.3.2. Regardless of this, the fact that the average MPI of all DMUs is greater than 1 (1.1359) indicates a trend in the ports' total productivity growth over the research period. There are three ports with the greatest efficiency gains: SP-10 (1.2277), SP-14 (1.2035), and SP-16 (1.3778). This can be explained by the great levels of technical and technological efficiency

attained by these DMUs. Figure 4 depicts the trend in productivity efficiency, which is nearly identical to the trend in the majority of DMUs when compared to the trend in Figure 2 (catch-up). In particular, SP-10 and SP-14 exhibit the highest volatility of all DMUs; both DMUs had a dramatic fall from 2018 to 2020 and are forecast to continue declining through 2020–2021, before gradually increasing over the next two years.

**Table 9.** Total productivity change for the period 2018–2023.

| Frontier | 2018–2019 | 2019–2020 | 2020–2021 | 2021–2022 | 2022–2023 | Average |
|---|---|---|---|---|---|---|
| SP-01 | 0.8353 | 1.0000 | 1.0261 | 1.0089 | 1.0008 | 0.9742 |
| SP-02 | 1.1161 | 1.0051 | 0.9974 | 0.9922 | 0.9993 | 1.0220 |
| SP-03 | 0.2964 | 2.2964 | 0.8718 | 1.0268 | 1.0023 | 1.0987 |
| SP-04 | 0.7491 | 1.0638 | 1.0107 | 1.0095 | 1.0008 | 0.9668 |
| SP-05 | 1.3704 | 0.9200 | 1.0055 | 0.9847 | 0.9986 | 1.0558 |
| SP-06 | 0.6049 | 0.7579 | 1.2510 | 1.0349 | 1.0030 | 0.9304 |
| SP-07 | 1.1071 | 0.8855 | 1.0412 | 1.0014 | 1.0001 | 1.0071 |
| SP-08 | 0.8048 | 1.6351 | 0.8285 | 0.9846 | 0.9986 | 1.0503 |
| SP-09 | 1.1287 | 1.2051 | 0.8988 | 0.9842 | 0.9986 | 1.0431 |
| SP-10 | 2.1237 | 1.1617 | 0.8775 | 0.9778 | 0.9980 | 1.2277 |
| SP-11 | 1.2133 | 0.9484 | 0.9907 | 0.9935 | 0.9994 | 1.0291 |
| SP-12 | 1.0135 | 0.9383 | 1.0293 | 1.0024 | 1.0002 | 0.9968 |
| SP-13 | 0.9993 | 0.9048 | 1.0516 | 1.0050 | 1.0004 | 0.9922 |
| SP-14 | 2.0435 | 1.1979 | 0.8341 | 0.9496 | 0.9923 | 1.2035 |
| SP-15 | 1.3779 | 1.1968 | 0.8677 | 0.9746 | 0.9977 | 1.0829 |
| SP-16 | 1.1583 | 3.3176 | 0.4888 | 0.9307 | 0.9937 | 1.3778 |
| SP-17 | 1.7278 | 0.9838 | 0.8374 | 0.9623 | 0.9969 | 1.1016 |
| SP-18 | 0.9796 | 1.0114 | 0.9969 | 1.0004 | 1.0000 | 0.9977 |
| Average | 1.1472 | 1.2461 | 0.9392 | 0.9902 | 0.9989 | 1.0643 |
| Max | 2.1237 | 3.3176 | 1.251 | 1.0349 | 1.003 | 1.3778 |
| Min | 0.2964 | 0.7579 | 0.4888 | 0.9307 | 0.9923 | 0.9304 |
| SD | 0.4642 | 0.6235 | 0.1546 | 0.0256 | 0.0027 | 0.1096 |

Note: calculated by the authors.

### 4.3.4. Comparative Analysis

The relation between the average technical change index and technological change indexes and the Malmquist production index of DMUs is depicted in Figure 5. Since the majority of ports performed much better in terms of technological development over the study period with 11 out of 18 DMUs achieving an average frontier-shift index greater than 1, the chart of technological change indexes between the DMUs demonstrates a stable chart in point of 1, while the average technical performance indicators of the DMUs fluctuated significantly in the whole period. Moreover, the MPI results from the technical change (catch-up index) and the technological change (frontier-shift index); the MPI chart has a similar pattern to the technical change chart, as illustrated in Figure 4. As a result, the evolution of each DMU's technical efficiency determines almost entirely the growth of its production change. This also explains why the patterns in Figure 4 for MPIs are nearly identical to the patterns in Figure 2 for all DMUs' catch-up indices. As port decision makers see how technological innovation has become the primary driver of the expansion and success of the port industry, they are increasing their focus on this area. On the other hand, to win the port industry, operators must make greater efforts to improve technical efficiency and production to optimize their capital, including labor, equipment and material suppliers, and investment.

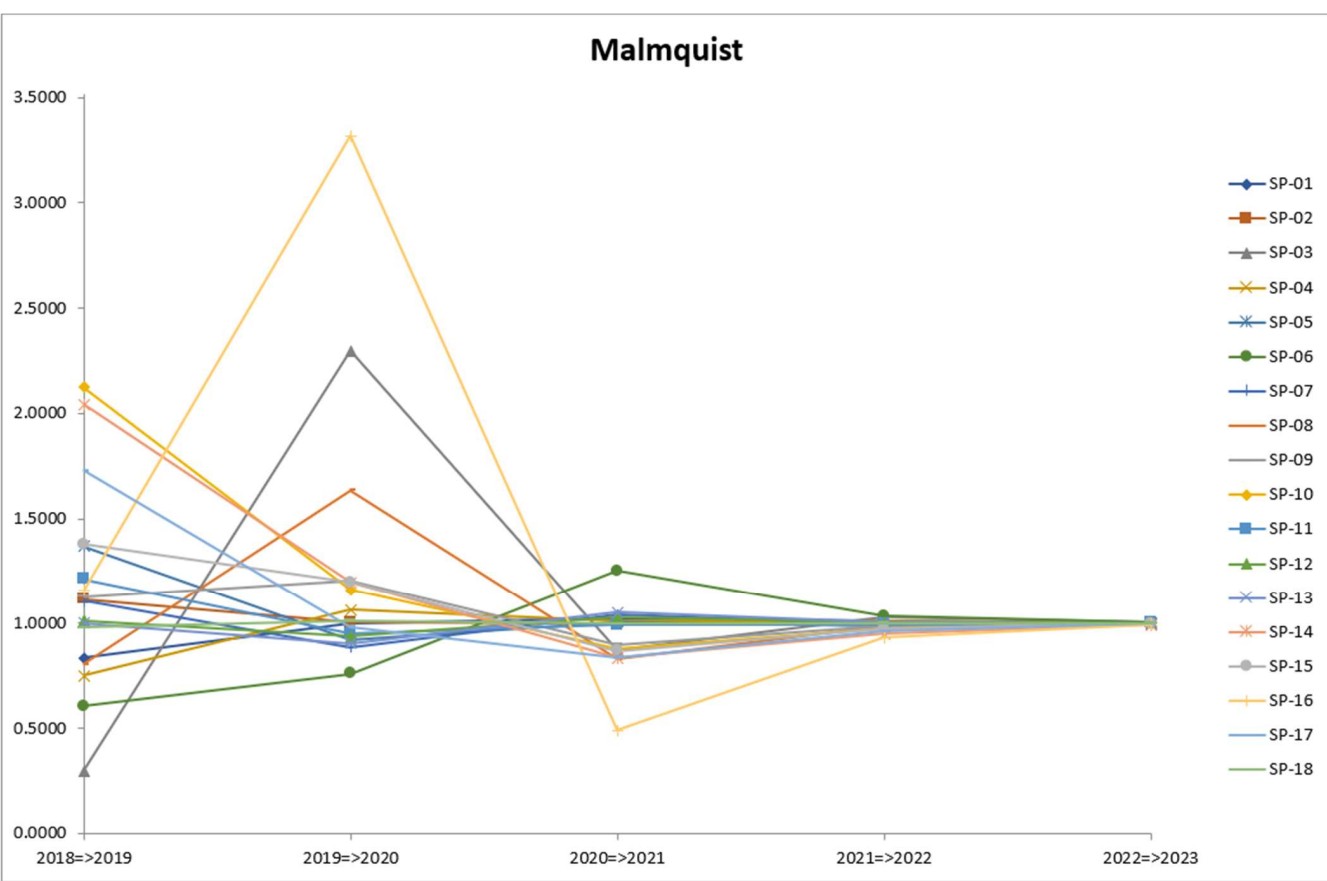

**Figure 4.** Total productivity change (Malmquist).

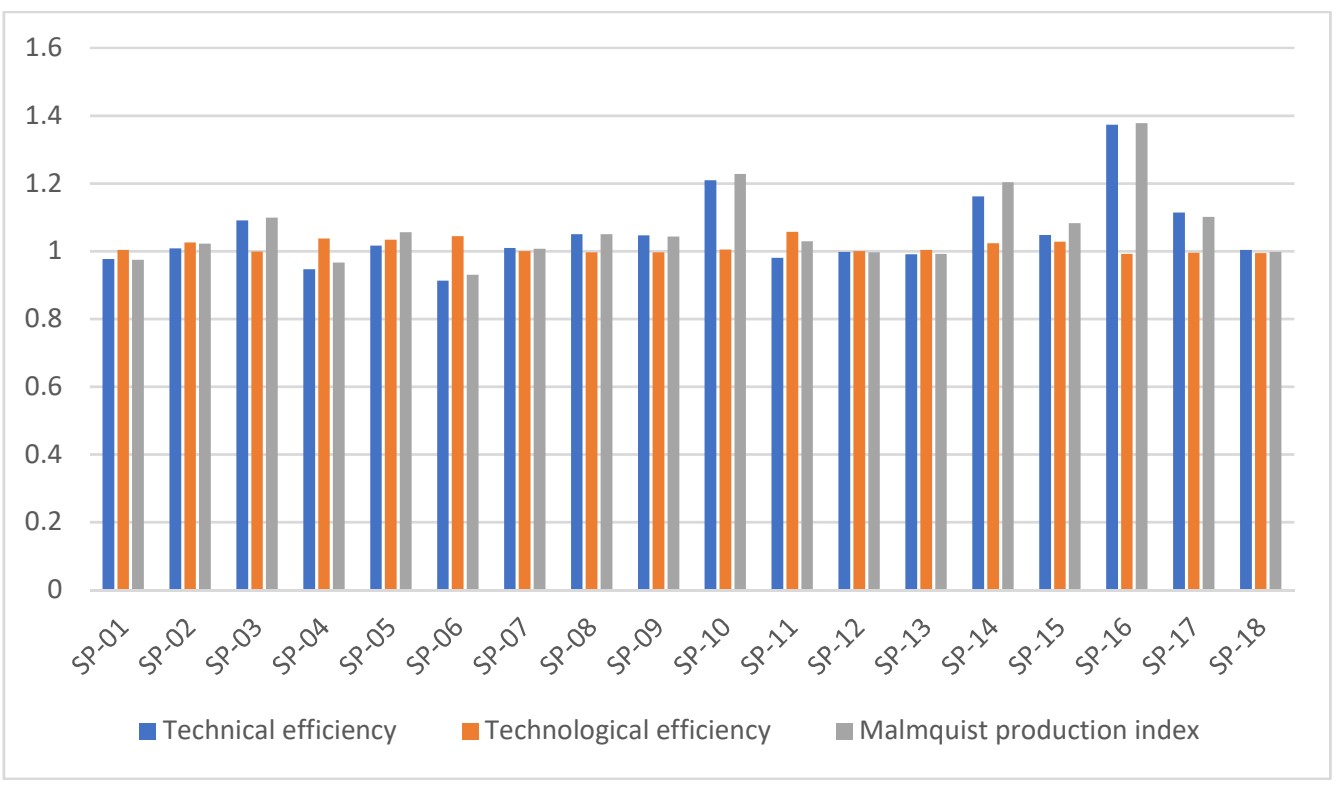

**Figure 5.** Comparison of average efficiency indexes.

*4.4. Discussions*

With a coastline of over 3260 km, Vietnam has enormous potential for seaport development. The largest ports in Vietnam are in the north and south, each with distinct advantages. Thus, assessing and forecasting the port industry's performance is viewed as a critical issue to ensure the port industry's long-term viability. As a result, this study analyzed and forecasted the performance of Vietnam's top 18 ports using a past–present–future model in DEA based on the resampling technique and the Malmquist model. The future efficiency was forecast using the resampling method in DEA using the past–present–future model. Due to the primary objective of forecasting accuracy, this stage considered the appropriate number of replicas and a valid prediction model. To ensure that the appropriate number of replicas was selected, 5000 and 500 replicas were created. These two tests produced statistically insignificantly low results, while the DEA scores were comparable. Thus, 500 replicas are available for this study. The final stage was to determine which of the three available prediction models, trend, Lucas weight, or hybrid, was the most accurate. The accuracy of these three models was determined by comparing the computed efficiency scores for each prediction model to the actual Super-SBM model scores. The ratio with the smallest value is the forecast-actual ratio. After comparison, the Lucas weight model was chosen as the prediction model with the smallest difference between the predicted and actual Super-SBM model scores. Finally, we forecasted the efficiency of these 18 ports using the Lucas weight prediction model.

Then, The DEA Malmquist productivity index was utilized to examine efficiency gains in the whole research period, 2018–2023. According to MPI's findings, 66.7% of the selected ports improved their production efficiency on average during the research period. The DEA Malmquist results for 2018 to 2020 show that most DMUs fluctuated dramatically, then are predicted to decrease modestly over the forecast period 2020–2023. In fact, due to the fourth outbreak and the high level of distancing measures implemented production, import, and export activities of enterprises were disrupted, especially when the epidemic broke out in an industrial zone and many enterprises could not meet the local working conditions, resulting in a serious impact on the output of goods [62]. However, there are three ports with the greatest efficiency gains in the entire research period, including SP-10 (Dong Nai), SP-14 (Lotus), and SP-16 (SSIT).

Regarding container cargo, most regions with a large volume of container cargo throughput experienced positive growth as Dong Nai, SSIT [63]. SP-16 (SSIT) achieved the highest productivity in 2019–2020, followed by a sharp decline in the next period. At the end of 2020, due to Vietnam's export demand and the high demand for empty containers, SSIT has had good growth in the volume of mother ships docked, and many ad-hoc ships call at lower port empty containers. Total mother ship volume at SSIT increased 134% TEU over the same period last year. The total throughput through the wharf will reach more than 1,000,000 TEU by 2020 [64]. Likewise, these ports experienced the highest Malmquist production efficiency; this can be explained by the fact that these DMUs achieved high technical and technological efficiency levels. Significantly, seaport organizations worldwide, particularly in Vietnam, continually upgrade technology to maximize port operations and communication. Technology modernizes ports by improving and simplifying trade movement, but it also reduces carbon footprint [65]. As a result, it can play a critical role in determining the competitiveness of seaport organizations.

## 5. Conclusions

Investing in the development of marine transport is critical for the economic development of a coastal country such as Vietnam. However, as with any nation, developing an appropriate strategy for seaport growth necessitates an assessment of current performance and a thorough examination of the causes of inefficiencies. Seaport productivity evaluation is a critical concept that focuses on resource utilization efficiency. This assessment can take various forms, the most common of which is efficiency measurement. When considering multiple inputs and outputs, DEA can be an effective tool for evaluating operational effi-

ciency. This study used the two-stage DEA model to examine port efficiency in Vietnam from 2018 to 2020 and forecasted the future performance of the country's top 18 seaports. As a result of this analysis, the following conclusions are drawn.

First, this research conducted a new comparative evaluation of the seaport using a combination of the DEA Malmquist and resampling models. The total efficiency of Vietnamese ports is not high due to limited technical and technological efficiency, and some Vietnamese ports are extremely inefficient (with efficiency scores far from the efficient frontier). This demonstrates that Vietnamese ports have had little impact on Vietnam's export competitiveness. Thus, Vietnamese ports have a significant opportunity to increase their operational efficiency and contribute to the country's international competitiveness and trade performance. Port efficiency can significantly impact Vietnam's overall export performance and economic development. Moreover, decision-makers must enhance their technical capabilities, equipment technology, growth strategies, and resource allocation for critical projects in order to maintain their competitive advantage, such as expanding port capacity in the most dynamic economic area for international competitiveness, trying to reform complex customs procedures, and prioritizing the development of infrastructure connecting ports to the territory.

Second, this study conducts a reliable forecast which is important for macroeconomic policymakers in setting policies. Moreover, forecasting port performance is a benefit for port administrators and investors to help them form strategic policies and correct their investment portfolios.

Third, the model's results will accurately reflect the current state of the seaport industry based on the performance of several successful seaport companies. As a result, our findings have important implications for helping seaport operators to better understand and determine critical port operations and development indicators. As a result, operators' technical and technological quality can be enhanced.

Fourthly, our findings and the analysis mechanism are expected to make significant academic and practical contributions. On the academic side, the paper fills a significant research gap relating to primary container terminals in Vietnam by systematizing research on container terminals in Vietnam. On the practical side, the study provides the Vietnam Government, policymakers, and terminal and port operators with detailed information about the performance of primary container terminals in Vietnam, allowing them to develop appropriate policies and strategies for improving performance.

However, the study has several limitations. Due to practical constraints, data could not be collected from all ports in Vietnam, limiting the dataset's scope to 18 major seaports. Second, because the study employs a DEA methodology, the results are relatively insensitive to the input and output variables chosen. Third, the panel data set spans only three years. As a result, future research should include additional input and output variables and broaden the scope of data collected in the terminal count and analysis period. Moreover, future research could be integrated with multiple criteria decision-making (MCDM) techniques [66,67] or machine learning techniques [68] to evaluate port capacity under an unclear environment.

**Author Contributions:** Conceptualization, C.-N.W. and P.-H.N.; Data curation, T.-L.N., T.-H.T. and H.-C.L.; Formal analysis, C.-N.W.; Funding acquisition, P.-H.N.; Investigation, T.-G.N., D.-T.N. and H.-T.P.; Methodology, T.-L.N. and P.-H.N.; Project administration, C.-N.W. and P.-H.N.; All authors have read and agreed to the published version of the manuscript.

**Funding:** This research was partly supported by Decision 1342/QD-DHFPT on 22 November 2021, from FPT University, Vietnam.

**Institutional Review Board Statement:** Not applicable.

**Informed Consent Statement:** Not applicable.

**Data Availability Statement:** Supplementary data to this article can be found online at https://data.mendeley.com/datasets/bhpmhm6pg2/2 (accessed on 11 April 2022).

**Acknowledgments:** The authors also appreciate the support from the National Kaoshiung University of Science and Technology, National Taipei University of Technology and the Ministry of Sciences and Technology in Taiwan.

**Conflicts of Interest:** The authors declare no conflict of interest.

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
