# Peer review of "A Two-Stage DEA Approach to Measure Operational Efficiency in Vietnam’s Port Industry"

_mathematics, doi:10.3390/math10091385_

Round 1

Reviewer 1 Report

The paper "A Two-Stage DEA Approach to Measure Operational Efficiency in Vietnam Port Industry

Aims to develop a data envelopment analysis (DEA) model to measure the performance efficiency of the 18 major seaports in Vietnam. It uses the DEA resampling technique (to predict future performance), and the DEA Malmquist model (to analyse efficiency improvement).

The work is certainly interesting and has taken a lot of time and effort. I would like to congratulate the authors.

However, some difficulties remain and in order to overcome them I propose to take into consideration the following remarks:

  1. Note that two paragraphs have the same number (4.2.3. Total Productivity Change and

4.2.3. Comparative analysis).

  1. Table 8, in fact it is Table 7 (Table 8. Technical efficiency changes for the period 2018-2023).
  2. A large part of the beginning of the introduction consists of generalities that are not necessarily directly related to the subject of the article.
  3. Fundamental concepts for this work were not raised either in the introduction or in the literature review (e.g. operational efficiency...)
  4. We suggest that the authors revise the introduction and literature review to isolate the following elements:

- Vietnam, port activity, history and perspectives

- A conceptual approach: (port performance, efficiency, technical efficiency and operational efficiency .....)

- DEA, DEAMalmquist and resampling method

  1. Specify the year for which Vietnam is ranked 80/139. ?? even in reference no. (9) the information is not clear.
  2. The authors state that "By 2020, the government intends for the country's seaports to 62 carry 200 million tons of cargo and double that number by 2030" the information is outdated (truncated) since the article is written in 2022.
  3. Avoid radical expressions like ".... Resampling and DEA Malmquist model has never been published" in favour of more flexible expressions that put your judgement into perspective...

Author Response

Thanks for your positive comments on the previous manuscript. As requested, we have revised and improved the manuscript carefully. Our reply to the comments is listed in the attachment.

Reviewer 2 Report

The paper sent for review falls within the area of ​​interest of the "Mathematics" journal. The authors make an interesting analysis using the Two-Stage DEA Approach. However, in the presented form, the article cannot be published. It is required to introduce both substantive and editorial corrections to the content of the study.
Notes for authors:
1. Table 1 - line 166 - last position in the table - activity not allowed! You cannot refer to what has not been published yet and does not exist in the scientific space.
2. Line 211-2121: "Numerous statistical tools were utilised to validate and corroborate the data collected" - please provide the statistical tools and devices with the authors used for the analysis - only the Pearson correlation coefficient is mentioned.
3. Please consider removing figures 2, 3 and 4 from the texts. The drawings are illegible. Moreover, they duplicate the data in Tables 6, 7 and 8.
4. Figure 5 - please consider making three separate drawings for the analysed indices on the same scale. It will certainly improve both the aesthetics of the text and the readability of the presented results.
5. Lines 465-473 - Conclusions section is a summary rather than a conclusion from the research content. They are very general, and in the presented form, they are practically possible to formulate based on the literature analysis without considering the authors' research. This fragment requires intensive improvement.
6. There are unacceptable self-citations of the study's authors in the references, e.g. 2, 17, 18 or 1, 54, 59. Please correct it.
7. The references do not contain a number of the latest publications (2021-2022) on the use of the DEA method to analyse the effectiveness of the operation of various systems. Please supplement.

Author Response

(The authors gave the same response as above.)

Round 2

Reviewer 1 Report

I think that a commendable effort has been made and that the authors have tried to provide answers to the various questions concerning the first version of this work. 

Reviewer 2 Report

Thank you very much. I am satisfied with the work done by the authors to improve the text and the answers provided. I recommend publication of the article after editorial corrections from the publisher. Congratulations!